# Performance of the Photosynthetic Apparatus under Glass with a Luminophore Modifying Red-To-Far-Red-Light Ratio—A Case Study

**DOI:** 10.3390/cells12111552

**Published:** 2023-06-05

**Authors:** Krzysztof M. Tokarz, Wojciech Makowski, Barbara Tokarz, Ewa Muszyńska, Zbigniew Gajewski, Stanisław Mazur, Edward Kunicki, Olgierd Jeremiasz, Piotr Sobik, Paweł Nowak, Karolina Miernicka, Kinga Mrzygłód, Piotr Rozpądek

**Affiliations:** 1Department of Botany, Physiology and Plant Protection, University of Agriculture in Krakow, al. 29 Listopada 54, 31-425 Kraków, Poland; 2Department of Botany, Institute of Biology, Warsaw University of Life Sciences, Nowoursynowska 159/37, 02-776 Warsaw, Poland; 3Department of Horticulture, University of Agriculture in Krakow, al. 29 Listopada 54, 31-425 Kraków, Poland; 4Helioenergia Sp. z o.o., ul. Rybnicka 68, 44-238 Czerwionka-Leszczyny, Poland; 5Institute of Metallurgy and Materials Science, Polish Academy of Sciences, ul. Reymonta 25, 30-059 Kraków, Poland; 6Malopolska Centre of Biotechnology, Jagiellonian University, Gronostajowa 7A, 30-387 Kraków, Poland

**Keywords:** antioxidant system, chl *a* fluorescence, chloroplast ultrastructure, CO_2_ carboxylation, *Lactuca sativa*, photosynthetic pigments, photosystem I, photosystem II, PSI and PSII acceptor side limitation

## Abstract

The aim of this study was to examine the effect of the modified light spectrum of glass containing red luminophore on the performance of the photosynthetic apparatus of two types of lettuce cultivated in soil in a greenhouse. Butterhead and iceberg lettuce were cultivated in two types of greenhouses: (1) covered with transparent glass (control) and (2) covered with glass containing red luminophore (red). After 4 weeks of culture, structural and functional changes in the photosynthetic apparatus were examined. The presented study indicated that the red luminophore used changed the sunlight spectrum, providing an adequate blue:red light ratio, while decreasing the red:far-red radiation ratio. In such light conditions, changes in the efficiency parameters of the photosynthetic apparatus, modifications in the chloroplast ultrastructure, and altered proportions of structural proteins forming the photosynthetic apparatus were observed. These changes led to a decrease of CO_2_ carboxylation efficiency in both examined lettuce types.

## 1. Introduction

Plant growth and development, as well as crop productivity in the natural environment are limited by various stress factors. Among these, one of the most-significant is sunlight [1,2]. Light is essential for photochemistry, which drives primary production. It is also the most-important environmental signal that modifies physiological processes and determines the course of plant morphogenesis [3,4]. Light for energy production is absorbed by photosynthetic pigments: chlorophylls—which absorb red and blue light most efficiently, and carotenoids, which absorb blue light [5]. Plants also evolved a sophisticated system of photoreceptors that perceive and assess the intensity, amount, duration, and direction, as well as the spectral composition of light and translate it into information necessary to optimize plant function [6]. Typically, photoreceptors include: phytochromes, which sense mainly red and far-red light, but also light from the blue and green spectrum; cryptochromes and phototropins, receptive to blue, green, and ultraviolet A; and a UV-B photoreceptor, sensing ultraviolet B [6]. Furthermore, plants perceive changes in light conditions via the photosynthetic apparatus and adjustments of its performance, thus modifying cellular metabolism [7]. Along with its role in energy production and signaling, light can also be harmful to plants, leading to photodamage due to imbalances in electron transport [8].

Both the functional and structural construction of the photosynthetic apparatus is dependent on changes in light quality and quantity [5]. Chlorophyll (Chl) and carotenoid (Car) pigments constitute the vast majority of external (light harvesting complexes (LHCs)) and internal (Cp47, Cp43) antenna complexes [9,10], transferring the collected energy to the reaction centers of photosystems II (PSII) and I (PSI) [11]. The absorbed energy triggers electron transport, which involves, beyond PSII and PSI, the cytochrome *b6f* (cyt *b6f*) protein complex and mobile transporters: plastoquinone (PQ), located in the thylakoid membrane, plastocyanin (PC), in the thylakoid lumen, and ferrodoxin (Fd), in the stroma [5,11]. Electron transport leads to the production of nicotinamide adenine dinucleotide phosphate (NADPH) and is accompanied by the transport of protons (H^+^) into the thylakoid lumen, which results in the synthesis of adenosine-3-phosphoric acid (ATP) [12]. These products are used in the Calvin–Benson cycle, which results in the assimilation of carbon from CO_2_ [13].

Light affects the formation, function, and regulation of the photosynthetic apparatus directly and indirectly [5]. Directly, light is perceived by antenna complexes. Moreover, directly, light is essential in the chlorophyll biosynthesis pathway, for the conversion of protochlorophyll to chlorophyll [14,15], while photoreceptors regulate the expression of nuclear genes encoding proteins involved in the synthesis of chlorophyll and carotenoid pigments, as well as the development and opening and closing of the stomata [5,6].

Photosynthetically active radiation (PAR) ranges between 400 and 700 nm [16]. Within this radiation, red light (600–700 nm), due to its lower photon energy content, appears to be the most-efficient to drive photosynthesis [17]. Red light, as perceived by phytochrome, plays a significant role in the formation of the photosynthetic apparatus, leaf morphogenesis, or carbohydrate accumulation [18]. However, it has been shown that the use of only red light as a light source leads to “red light syndrome”, manifested by impaired plant growth and development, especially of the leaves, and a decrease in photosynthetic efficiency [19,20]. Blue light (400–500 nm), which, through cryptochrome and phototropin, has been shown to regulate photosynthetic rates, chlorophyll formation, and stomatal opening [19,21,22], is necessary to avoid these symptoms. Thus, it seems that, for proper growth and development, plants require an appropriate ratio of blue and red light to provide optimal conditions for photosynthesis and photomorphogenesis. Following the above findings, to intensify crop production, light-emitting diode (LED) lamps are used, allowing the application of selected, narrow ranges of radiation for maximizing the efficiency of the photosynthetic apparatus. In this mode of light application, it is assumed that different wavelengths affect photosynthesis independently and additively, while the synergistic effects of different wavelengths are largely disregarded [23]. Indeed, it is increasingly apparent that, in addition to blue and red radiation, the overlooked green, far-red, and even UV radiation play a role in efficient photosynthesis and plant productivity [5,23,24].

Lettuce (*Lactuca sativa* L. var. *capitata*), with its short growth cycle, low energy requirements, and stable yield, is often exploited as a model crop to study plant responses to stress conditions, especially light and temperature [25,26]. Furthermore, lettuce is a top leafy vegetable cultivated in greenhouses [25,26,27].

The aim of this study was to examine the effect of the modified light spectrum (Figure 1) of glass containing red luminophore (PL Patent No. 240248 B1) [28] on the performance of the photosynthetic apparatus of lettuce cultivated in the soil in a greenhouse. Furthermore, such a luminophore is employed to produce glass with integrated photovoltaic panels, which can be used in the future to build greenhouses. With the application of a photoluminescent pigment, part of the absorbed energy through the fiber-optic effect would be transferred to the solar panels and converted into electricity [28]. However, the addition of luminophore to the glass alters its optical properties, and therefore, it is necessary to evaluate the plants’ response to the light conditions under such glass. Two types of lettuce were chosen for the study to see if the photosynthetic apparatus response mechanism to the applied conditions would be universal among closely related plants.

Our results showed that the performance of the photosynthetic apparatus of both types of lettuce cultivated under glass with red luminophore differed significantly compared to the control conditions. The study indicated that the red luminophore used changed the sunlight spectrum, providing an adequate blue:red light ratio, while a decreased red:far-red radiation ratio. Regarding the lettuce types tested, such light conditions led to a decrease in CO_2_ carboxylation efficiency, resulting from the disruption of linear electron transport due to a limitation on the acceptor side of PSII and PSI.

## 2. Materials and Methods

### 2.1. Plant Material

The experimental material comprised two types of head lettuce (*Lactuca sativa* var. *capitata*): butterhead and iceberg. Seeds of the butterhead-type cultivar Melodion were purchased from Enza Zaden Ltd. (Warsaw, Poland), and seeds of the iceberg cultivar from Elenas from Rijk Zwaan Ltd. (Blonie, Poland).

### 2.2. Cultivation Conditions

The experiments were conducted in 2020 at the University of Agriculture in Kraków (Poland) in the high-tech greenhouse of the Faculty of Biotechnology and Horticulture (transplants’ production; 50°03′ N, 19°57′ E) and in small, temporary greenhouses located at the vegetable experimental station (main experiment; 50°08′ N, 19°85′ E). Lettuce seeds were germinated in 96-cell multi-pots (60 × 40 cm) filled with Florabalt Seed (Floragard Vertriebs GmbH, Oldenburg, Germany) (pH 5.6; N 140, P_2_O_5_ 80, and K_2_O 190 mg·L^−1^) and kept in greenhouse conditions for 5 weeks. Before transplanting, the soil was fertilized with the fertilizer YaraMila Complex (5% N-NO_3_, 7% N-NH_4_); P—11% P_2_O_5_; K—18% K_2_O; Mg—2.7% MgO; S—20% SO_3_; B—0.015%; Fe—0.20%, Mn—0.02%; Zn 0.02%) (Yara Poland Ltd., Szczecin, Poland) at a dose of 45 g·m^−2^ (450 kg·ha^−1^). Five-week-old seedlings were transplanted to soil in two types of greenhouses: (1) covered with transparent glass (control) and (2) covered with glass containing red luminophore (red) (Helioenergia Ltd., Czerwionka-Leszczyny, Poland) [28]. The spectral characteristics of the transmitted sunlight, measured with a SpectraPen mini (Photon Systems Instruments, Drásov, Czech Republic) after dark calibration, are presented at Figure 1. Two weeks after transplantation, the plants were topdressed with amonium sulphate (34% N) at a dose of 5 g·m^−2^ (50 kg·ha^−1^). Plants were irrigated 2 times in 2-week intervals (first, just after transplanting), each time at a dose of 20 mm and mechanically weeded 3 times.

### 2.3. Evaluation of Photosynthetic Apparatus Performance

The photosynthetic apparatus performance was evaluated on living plants after four weeks of cultivation.

#### 2.3.1. Photosynthetic Pigment Concentration Assessment

Individual leaves were collected from plants on which chlorophyll *a* fluorescence and gas exchange were measured. Photosynthetic pigment content analyses were conducted according to the spectrophotometric method of Wellburn [29]. Immediately after collection, leaves were weighted and homogenized with 80% acetone (30 mL) in ice-cold conditions. Samples were centrifuged for 15 min at 4800× *g* at 4 °C. Diluted extracts were measured at 470, 646, and 663 nm, which correspond to chlorophyll *a*, chlorophyll *b*, and carotenoids’ absorbance, respectively. The absorbance of samples was measured using the double-beam spectrophotometer U-2900 (Hitachi High-Technologies Corporation, Tokyo, Japan). The content of the photosynthetic pigments were calculated using equations. Moreover, total chlorophylls (Chl *a* + *b*) and the ratios of the pigments (Chl *a*/*b*) were calculated.

#### 2.3.2. Chl *a* Fluorescence Measurements

The chlorophyll *a* fluorescence induction kinetics were recorded using the Handy-PEA (Hansatech, King’s Lynn, UK) spectrofluorometer according to standard procedures. Individual leaves (young and fully developed) of ten randomly chosen plants of each cultivar and treatment were dark-adapted for approximately 25 min. After dark adaptation, the leaves were subjected to a saturating flash of red light (λmax = 650 nm, 2500 μmol (quants) m^−2^ s^−1^). Standard procedures led to extract parameters such as: F0, Fm, F50μs, F100μs, F300μs, F2ms, F30ms, and Area. Moreover, according to the formulas of Strasser et al. [30], Jiang et al. [31], Kalaji et al. [32], and Goltsev et al. [33], selected structural and functional parameters were calculated. These were: Fv, Fv/Fm, Fv/F0, VJ, VI, Sm, φPo, φEo, ψEo, ρRo, δRo, φRo, ABS/RC, TRo/RC, ETo/RC, DIo/RC, ABS/CSo, TRo/CSo, ETo/CSo, DIo/CSo, RC/CSo. A description of the parameters is presented in the Abbreviations.

#### 2.3.3. Gas Exchange Measurements

The measurement of gas exchange was performed on individual leaves of three randomly chosen plants of each cultivar and treatment using the LCpro-SD (ADC BioScientific Ltd., Hoddesdon, UK)—a portable gas-exchange system with a measuring chamber. The conditions, within the cuvette, used to measure net photosynthesis (Pn), stomatal conductance (Gs), transpiration rate (E), and the intercellular concentration of CO_2_ (Ci), were: CO_2_-saturated conditions (650 μmol·mol^–1^), air flow of 300 mol·s^–1^, relative humidity of 50–55%, organ temperature of 25 °C, and red light intensity of 300 μmol (quanta) m^−2^·s^−1^. The photosynthetic response curves to light were performed on the same plants used for net photosynthesis for progressively reducing photosynthetically active radiation (PAR) intensities, ranging from 1500 to 0 µmol (quanta) m^−2^·s^−1^ (in steps of 300, 100, 50, 20, 0, 100, 300, 500, 1000, 1500, and 300 µmol (quanta) m^−2^·s^−1^). Leaves were adapted to each light intensity for 2, 2, 2, 2, 2, 2, 2, 2, 2, 2, and 7 min, respectively, before recording the data points.

#### 2.3.4. Structural and Functional Photosynthetic Protein Content Determination

The Western blot technique was used to determine the presence and abundance of selected proteins. Proteins were extracted from chloroplasts isolated from the lettuce leaves. The isolation of chloroplasts was conducted according to the method described by Tokarz et al. [10]. Freeze-dried chloroplasts were extracted according to the method of Laureau et al. [34] with modifications using an extraction buffer composed of 100 mM Tris-HCl, pH 8.0, 10% sucrose, 0.2% β-mercaptoethanol (β-ME), and 2% polyvinylpolypyrrolidone (PVPP). The protein concentration was assessed according to Bradford [35]. A calibration curve was prepared using BSA as the standard. Before electrophoresis, gels were warmed up to room temperature. SDS-PAGE electrophoresis was performed on 12% polyacrylamide gels (Bio-Rad, Hercules, CA, USA) at 4 °C at 30 mA for 15 min and 20 mA for 90 min using a vertical gel electrophoresis system (Mini-PROTEAN^®^ Tetra Vertical Electrophoresis Cell, Bio-Rad). A loading buffer composed of 62.5 mM Tris-HCl, pH 6.8, 2% SDS, 25% glicerol, 0.01% bromophenol blue, and 5% β-ME was used. Following electrophoresis, proteins from the polyacrylamide gel were electroblotted onto a nitrocellulose membrane (pore size 0.2 µm) using a semi-dry electroblotter (Trans-Blot Turbo Transfer System, Bio-Rad, CA, USA). A buffer containing 48 mM Tris (pH 9.2), 39 mM glycine, 20% methanol, and 1.3 mM SDS was used for transfer. The parameters of transfer were: room temperature, 25 V (limiting parameter), and 0.5–0.7 A for 7–8-min depending on the protein mass. TBST buffer (0.12 M Tris-HCl, pH 8.0, 8.8% NaCl, 0.5% Tween 20) containing 3% dry milk was used for blocking the membranes at room temperature for 2 h. The membranes were then probed with rabbit primary antibody (Ab) against the PSI-A core protein of photosystem I (PsaA, AS06 172, Agrisera, Vinnas, Sweden), PSI-B core subunit of photosystem I (PsaB, AS10 695, Agrisera), CP47 protein of PSII (PsbB, AS04 038, Agrisera), CP43 protein of PSII (PsbC, AS11 1787, Agrisera), 33 kDa of the oxygen-evolving complex (OEC) of PSII (anti-protein) (PsbO, AS06 142-33, Agrisera), 16 kDa protein of OEC of PSII (PsbQ, AS06 142-16, Agrisera), PSI type I chlorophyll-*a*/*b*-binding protein (Lhca1, AS01 005, Agrisera), LHCII type I chlorophyll-*a*/*b*-binding protein (Lhcb1, AS01 004, Agrisera), RuBisCo activase (RA, AS10 700, Agrisera), RuBisCo large subunit (RbcL, AS03 037, Agrisera), and catalase (CAT, AS09 501, Agrisera). Membranes were washed with TBST buffer and incubated with a horseradish-peroxidase-conjugated anti-rabbit secondary antibody (HRP, AS09 602, Agrisera) at a dilution of 1:10,000 in TBST buffer for 1.5 h. After washing with TBST buffer, a solution of 5-bromo-4-chloro-3-indolyl phosphate (BCIP) and nitro blue tetrazolium (NBT), made in a buffer composed of 100 mM Tris (pH 9.5), 100 mM NaCl, and 5 mM MgCl_2_, was used to detect antigen–antibody complexes. The membranes were scanned with an Epson Perfection V750 Pro scanner. ImageJ software (Version 1.53k, open-source software, NIH, Bethesda, MD, USA) was used for the densitometry analysis of the scanned membranes. The content of each protein is given in arbitrary units, defined as the area under the curve. Area values were calculated in reference to the mean area value for the control on each gel expressed as 1.

#### 2.3.5. Transmission Electron Microscopy Observation

After 4 weeks of cultivation, fragments of leaves were collected. The material was fixed in 2% paraformaldehyde and 2% glutaraldehyde in 0.1 M cacodyl buffer (pH 7.2) for 2 h. The sections were then washed four times in cacodyl buffer and fixed in a solution of 2% osmium tetroxide in cacodyl buffer for 3 h at 4 °C. After this time, the material was dehydrated through a stepwise ethanol series and replaced with propylene oxide, then embedded in glycidyl ether 100 epoxy resin (SERVA, Heidelberg, Germany). Resin polymerization was carried out at 65 °C for 24 h. Semi-thin sections were prepared with a Jung RM 2065 (Leica, Wetzlar, Germany) microtome, stained with methylene blue and azure B and examined under a light microscope (Olympus-Provis, Tokyo, Japan). Ultra-thin sections were prepared with a Leica Ultracut UCT microtome (Leica, Wetzlar, Germany), collected on formvar-coated grids and stained with uranyl acetate, followed by lead citrate for 1 min. The examination was performed in a transmission electron microscope (Morgagni 268D, Hillsboro, OR, USA). Additionally, on ten randomly chosen chloroplasts of each lettuce type and condition, we measured the chloroplast size–length (using scale bars) and counted the number of grana, starch grains, and plastoglobuli.

### 2.4. Evaluation of Leaf Antioxidant Activity

#### 2.4.1. Guaiacol Peroxidase Activity Evaluation

The guaiacol peroxidase (POD) activity was assayed according to Lűck [36] as follows: Leaf samples (2 g) were homogenized in an ice bath (4 °C) in 10 mL of 50 mM potassium phosphate buffer (pH 6.2). The mixture was centrifuged at 13.968× *g* for 15 min at 4 °C. Then, 2 mL of plant extracts diluted five times was mixed with 2 mL of potassium phosphate buffer and 0.2 mL of a 1% solution of p-phenylenediamine. The peroxidase activity was assessed by measuring absorbance at 485 nm on a UV–VIS Helios Beta spectrophotometer (Spectronic Unicam, Cambridge, UK) one minute and two minutes after the addition of 0.2 mL 0.1% H_2_O_2_ to each sample. A blind sample was prepared as described above, but without H_2_O_2_ addition. A unit of enzyme activity (U) is expressed as an increase in absorbance of 0.1 per minute.

#### 2.4.2. Glutathione Content Evaluation

The reduced form of glutathione (GSH) was assayed using the method described by Guri [37], with some modifications. For this, 2 g of fresh leaves were chopped and homogenized with 10.0 mL of 0.5 mM EDTA and 3% trichloroacetic acid (TCA) in an ice bath (4 °C). The extract was centrifuged at 13.968× *g*, for 10 min, at 4 °C. The supernatant (2 mL) was mixed with 5 mL K-phosphate buffer (pH = 7.0) to bring the solution pH to the value of ca. 7.0. Next, 1 mL of K-phosphate buffer and 0.1 mL Ellman’s reagent (5,5-dithiobis-2-nitrobenzoic acid (DTNB)) (Merck KGaA, Darmstadt, Germany) were added to 2 mL of this mixture. The content of reduced glutathione was assessed by measuring absorbance at 412 nm on a UV–VIS Helios Beta spectrophotometer, against a blind sample, prepared as described above, but with 1.1 mL K-phosphate buffer and without Ellman’s reagent. The GSH content was calculated based on the calibration curve of GSH and expressed in mg per 1 g fresh weight (FW).

### 2.5. Statistical Analyses

STATISTICA 12.0 (StatSoft Inc., Tulsa, OK, USA) was used to perform the statistical analyses. The results, within each parameter and lettuce type, were subjected to one-way analysis of variance (ANOVA). The Duncan post hoc test at *p* ≤ 0.05 was used to determine the significance of the differences between the means. All of the spectrophotometric determinations were made in five replications. Chl *a* fluorescence measurements were performed in ten replications. Gas exchange measurements and electrophoresis were performed in three replications.

## 3. Results

### 3.1. Photosynthetic Apparatus Response to Red Light

#### 3.1.1. Photosynthetic Pigment Concentration

To evaluate changes in the antennae of the lettuce photosynthetic apparatus cultivated in the red glasshouse, the content and ratio of the photosynthetic pigments were analyzed. After 4 weeks of cultivation, there were no changes in the concentration of the chlorophylls (Chl *a*, Chl *b*, Chl *a* + *b*) and carotenoids (Car) in iceberg lettuce leaves in the red glasshouse in comparison to the control (Table 1). Similarly, no change in the Chl *a*/*b* ratio was found (Table 1), whereas, in the leaves of butterhead lettuce, a significant increase of the chlorophylls’ and no change in the carotenoids’ concentration were observed in the red glasshouse compared to the control. However, the ratio of Chl *a*/*b* increased in the leaves of plants grown in the red glasshouse (Table 1).

#### 3.1.2. Chl *a* Fluorescence

Chl *a* fluorescence was measured to describe the efficiency of PSII photochemistry. The values of the measured and calculated PSII parameters were normalized against the control (set as 1) and presented on radar charts (Figure 2a,b). Raw values of these parameters are presented in the Appendix A. The fluorescence parameters (minimum (F_0_), maximum (Fm), and variable (Fv) fluorescence) decreased significantly in both types of lettuce examined grown in the red glasshouse compared to the control (Figure 2a,b, Appendix A). The maximum quantum yield of PSII (Fv/Fm) and the activity of the water-splitting complex (Fv/F_0_) did not change in the red glasshouse in both types of lettuce (Figure 2a,b, Appendix A). The relative variable fluorescence at 2 ms (V_J_) and relative variable fluorescence at 30 ms (V_I_) increased significantly in both types of lettuce examined cultivated in the red glasshouse (Figure 2a,b, Appendix A). The reduced plastoquinone pool (Area) and total electron carriers per reaction center (RC) (Sm) decreased in both lettuce types in the red glasshouse (Figure 2a,b, Appendix A). The parameters describing yield or flux ratios (φPo, φEo, φRo, δRo, ρRo, and ψEo) decreased significantly in the tested plants cultivated in the red glasshouse (Figure 2a,b, Appendix A). Specific fluxes or activities per RC (ABS/RC, TRo/RC, ETo/RC, and DIo/RC) increased significantly in butterhead and iceberg lettuce (Figure 2a,b, Appendix A). The trapped energy flux per cross-section (CS) (TRo/CSo) and electron transport flux per CS (ETo/CSo) decreased, while the dissipated energy flux per CS (DIo/CSo) increased in both butterhead and iceberg lettuce (Figure 2a,b, Appendix A), whereas, the amount of active PSII RCs per CS (RC/CSo) decreased significantly also in both examined lettuce types cultivated in the red glasshouse in comparison to the control glasshouse (Figure 2a,b, Appendix A).

#### 3.1.3. Gas Exchange

Measurements of gas exchange using infrared were carried out to determine the photosynthetic efficiency of the examined plants. Net photosynthesis (Pn) decreased significantly in both tested lettuce types in the red glasshouse compared to the control (Figure 3a,c). In turn, transpiration (E) and stomatal conductance (Gs) decreased significantly only in iceberg lettuce (Figure 3b), while the intercellular CO_2_ concentration (Ci) did not change either in butterhead or iceberg lettuce (Figure 3a,c). Moreover, the leaf photosynthesis efficiency of iceberg lettuce growing in the red glasshouse was significantly lower in low light intensity (0–50 μmol·m^−2^·s^−1^) and in the range of light intensity between moderate (100 μmol quanta m^−2^·s^−1^) and high (1500 μmol quanta·m^−2^·s^−1^) (Figure 3d), while the leaf photosynthesis efficiency of butterhead lettuce, cultivated in the same conditions, was lower in light intensity between 100 and 1500 μmol quanta·m^−2^·s^−1^ (Figure 3b).

#### 3.1.4. Structural and Functional Photosynthetic Proteins

The quantitative participation of photosystem I (PsaA, PsaB, Lhca1), photosystem II (PsbB, PsbC, PsbO, PsbQ, and Lhcb1), proteins, RuBisCo (RbcL), and RuBisCo activase (RA) was estimated by SDS-PAGE and immunoblotting in lettuce isolated chloroplasts. The content of PsaA and PsaB, core proteins of photosystem I, increased both in butterhead and iceberg lettuce types growing in the red glasshouse (Figure 4a,b). The Lhca1 content increased in iceberg lettuce chloroplasts, but did not change in butterhead ones (Figure 4a,b). In turn, the content of the core antenna of PSII, PsbB, increased in both lettuce types cultivated in the red glasshouse, whereas PsbC content did not change either in butterhead or iceberg chloroplasts (Figure 4a,b). Similarly, the content of Lhcb1, the LHCII type I chlorophyll *a*/*b*-binding protein, did not change in either lettuce type cultivated in the red glasshouse (Figure 4a,b). Moreover, the content of the subunits constituting the oxygen evolving complex (PsbO and PsbQ) did not change in butterhead and iceberg lettuce chloroplasts (Figure 4a,b). In contrast, a decrease in the content of RuBisCo activase (RA) was recorded in both lettuce types’ chloroplasts (Figure 4a,b), while the content of the RuBisCo large subunit (RbcL) decreased only in butterhead lettuce chloroplasts in the red glasshouse (Figure 4a).

#### 3.1.5. Chloroplast Ultrastructure

The TEM observation revealed the ultrastructure of chloroplasts from the leaves of butterhead and iceberg lettuce cultivated in the control and red glasshouses (Figure 5, Appendix A). Significant differences were observed in the ultrastructure of butterhead and iceberg lettuce chloroplasts from plants cultivated in the red glasshouse in comparison to the control (Appendix A). Under control conditions, the chloroplasts had a regular shape, numerous grana (Appendix A), and a clustered arrangement of thylakoids (Figure 5a,b,e,f, Appendix A). There were no differences in the size (length) of butterhead chloroplasts from plants cultivated in control and red glasshouse conditions (Appendix A). However, iceberg chloroplasts of plants from the red glasshouse were significantly larger (longer) than the chloroplasts of plants from the control conditions (Appendix A). Moreover, we noted also numerous plastoglobuli (Figure 5a,b,e,f, Appendix A) and, in butterhead lettuce, visible starch grains (Figure 5a,b, Appendix A). In the red glasshouse conditions, chloroplasts had less numerous grana and plastoglobuli (Figure 5c,d,g,h, Appendix A). In iceberg lettuce chloroplasts, additionally poorly visible grana, a looser arrangement of thylakoids, and practically no starch grains were observed (Figure 5g,h, Appendix A).

## 4. Discussion

According to the available literature, an optimally balanced ratio of blue to red light (1:1–1:7) significantly improves the photosynthetic capacity of leaves [38,39]. In the presented study, the ratio of blue to red light (660 nm:450 nm) was approximately 1:1 in the control conditions and 1:2 under glass with red luminophore (Figure 1a,b). Despite the correct ratio of red and blue light, abnormalities in the structure and function of the photosynthetic apparatus were observed in both examined lettuce types cultivated in the red glasshouse. These differences may be explained by the influence of other factors, including the other aspects of the spectral composition of the light used. The relevance of a reasonably high blue:red light ratio is the subject of numerous studies [4,19,21,22,40,41]. In contrast, the contribution and impact of far-red radiation and its ratio to red light is very often neglected. Most studies dealing with this topic focus on the far-red and red-light-induced photomorphogenic response associated with phytochrome induction [24]. Among studies that focus on the structure and function of the photosynthetic apparatus, there is a considerable discrepancy in the description of plant responses to the ratio of red to far-red [24,42,43]. In our study, the red:far-red (660 nm:720 nm) ratio in red glasshouse was approximately 1:0.6 in comparison to 1:0.8 in the control glasshouse. Few studies indicate that a low red:far-red ratio leads to a reduction in the number of grana thylakoids and their stacking degree in the ultrastructure of chloroplasts [24]. Our results showed that the butterhead lettuce chloroplast ultrastructure was less susceptible to a changed spectral composition than iceberg lettuce, which were significantly longer than the control ones (Appendix A). In contrast, more far-red light, in tobacco studies, caused chloroplast elongation [44]. However, chloroplasts of both lettuce types in the red glasshouse had significantly less grana, starch grains, and plastoglobuli (Appendix A). Meanwhile, studies of other authors indicated a very different response of the chloroplast ultrastructure (decrease or increase of grana thylakoids) to excessive or deficient far-red light [24,44,45].

Chloroplast ultrastructural changes arise from alterations in the structure of the photosynthetic apparatus, of which protein complexes (e.g., PSII, PSI, LHCII, LHCI) are located in the membranes, building the grana and stroma thylakoids [11,46]. In this study, under enhanced red radiation, an increase in chlorophyll pigments was observed in butterhead lettuce accompanied by an increase in the content of the cortical subunits of PSI, i.e., PsaA and PsaB (Figure 4a), which are responsible for light harvesting, charge separation, and electron transport [47]. Indeed, only Chl *a* is found in photosystems [11], so an increase in the Chl *a*/*b* ratio (Table 1) is associated with an increased proportion of photosystem proteins relative to antenna proteins, as observed in butterhead lettuce (Figure 4a). In contrast, iceberg lettuce showed no change in pigment content, but an increase in PsaA, PsaB, and Lhca1—*a* chlorophyll *a*/*b*-binding protein of LHC of PSI (Figure 4b). Since PSI is localized in the stroma thylakoids, an increase in the content of these subunits is associated with a change in chloroplast structure, where the proportion of stroma thylakoids increases and the stacking of grana thylakoids decreases [11].

The structural modifications described above were accompanied by changes in the function and efficiency of the photosynthetic apparatus. Although in plants grown in the red greenhouse, no electron limitation was observed on the donor side of PSII—no change in OEC capacity and structure, as evidenced by unchanged PsbO and PsbQ content (Figure 4) and Fv/F_0_ (Figure 2)—there was a reduction in the number of open reaction centers (RC) of PSII among all active reaction centers (V_J_ increase; Figure 2). A decrease in the efficiency of electron transporters (Area decrease; Figure 2) was also observed. There was a reduction in the content of the membrane PQ pool (Sm decrease; Figure 2) and, within this pool, a decrease in its rapidly reducing fraction (V_I_ increase; Figure 2), as well as the reserve PQ pool accumulated in plastoglobuli, the number of which decreased (Appendix A). Moreover, in both types of lettuce in the red greenhouse, a limitation in electron transport was observed both on the acceptor side of PSII, between Q_A_ and Q_B_ (φEo decrease; Figure 2) and on the acceptor side of PSI (ρRo, δRo, φRo decrease; Figure 2). The limitation on the acceptor side of PSI may result from the efficiency of the Calvin–Benson cycle, especially the amount and activity of RuBisCo [33]. A high proportion of far-red light activates RuBisCo [48]. On the other hand, the amount of RuBisCo also depends on light intensity, not only on light quality [49]. In addition, RuBisCo activity is regulated by the stroma pH, which is too low when electron transport is not efficient [50]. In our study, we observed both a decrease in activity (RA decrease; Figure 4) and in the amount of RuBisCo (RbcL decrease; Figure 4). All this led to a decrease in CO_2_ carboxylation efficiency (Pn decrease; Figure 3). The limitation on the acceptor side of PSII and PSI in the absence of a limitation on the donor side of PSII suggested the elevated generation of reactive oxygen species around PSII, as evidenced by the catalase content increase (Figure 6) and, in iceberg lettuce, also by the higher glutathione content (Figure 6).

Regarding the functioning of the photosynthetic apparatus, some researchers point to an increased linear electron transport and photosynthetic intensity in a reduced red:far-red ratio [51]. The meaning of far-red light is related to the enhanced oxidation of photosystem I, which, in combination with an efficiently functioning NADH thioredoxin reductase (NTRC), leads to the oxidation of plastocyanin and ferrodoxin. Furthermore, far-red radiation induces cyclic electron transport around PSI via both the PGR5 and PGRL pathways and NDH [52]. As a result, it enables efficient linear electron transport, as well as cyclic transport around PSI [52]. In our study, the light conditions in the red glasshouse were characterized by increased in the red:far-red ratio relative to the control, which resulted in the aforementioned changes in chloroplast ultrastructure and also the described limitation on the acceptor side of PSI, resulting from the lack of efficient PSI oxidation, consequently leading to the disruption of linear electron transport.

## 5. Conclusions

Our studies indicated that the ultrastructure and function of the photosynthetic apparatus of both lettuce types studied is more significantly affected by the respective red:far-red ratio than by the blue:red ratio. Despite the high value of the blue:red ratio, a low red:far-red ratio implies a decrease in the intensity of CO_2_ carboxylation, resulting from the disruption of linear electron transport due to the limitation on the acceptor side of PSII and PSI. The study indicated that the red luminophore used provides an adequate blue:red ratio, while a decreased red:far-red ratio. The disruption of photosynthetic efficiency observed in lettuce in our experiments may not necessarily be observed in other species grown under such light conditions, which requires further research.

## Figures and Tables

**Figure 1 cells-12-01552-f001:**
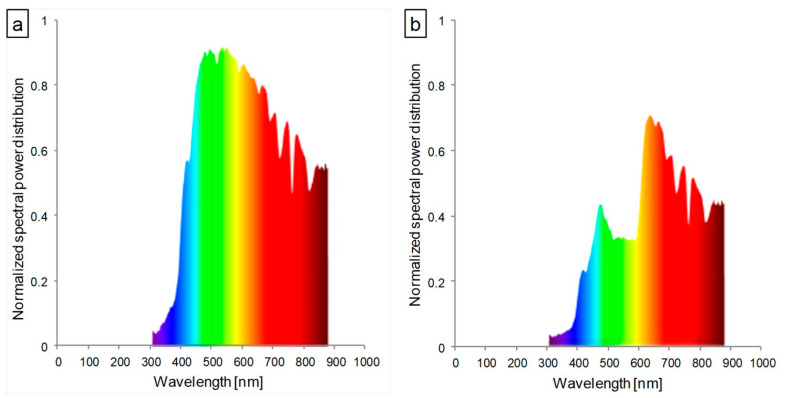
Spectral characteristics of the light transmitted through (**a**) transparent glass (control) and (**b**) glass containing red luminophore (red).

**Figure 2 cells-12-01552-f002:**
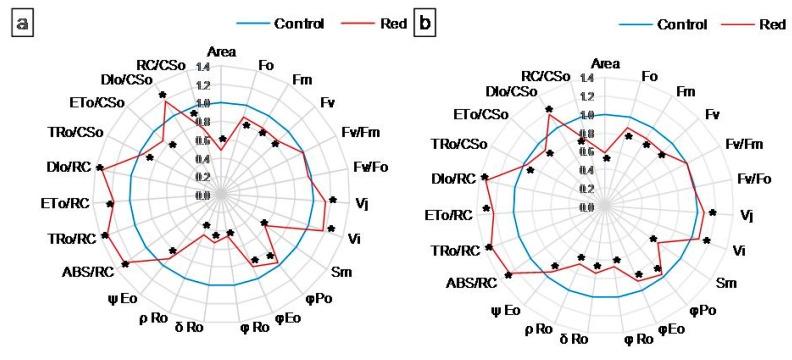
Structural and functional parameters of photosynthetic apparatus of (**a**) butterhead and (**b**) iceberg lettuce types cultivated in transparent (control) and red glasshouses; * statistically significant difference within each parameter at *p* ≤ 0.05; (*n* = 10).

**Figure 3 cells-12-01552-f003:**
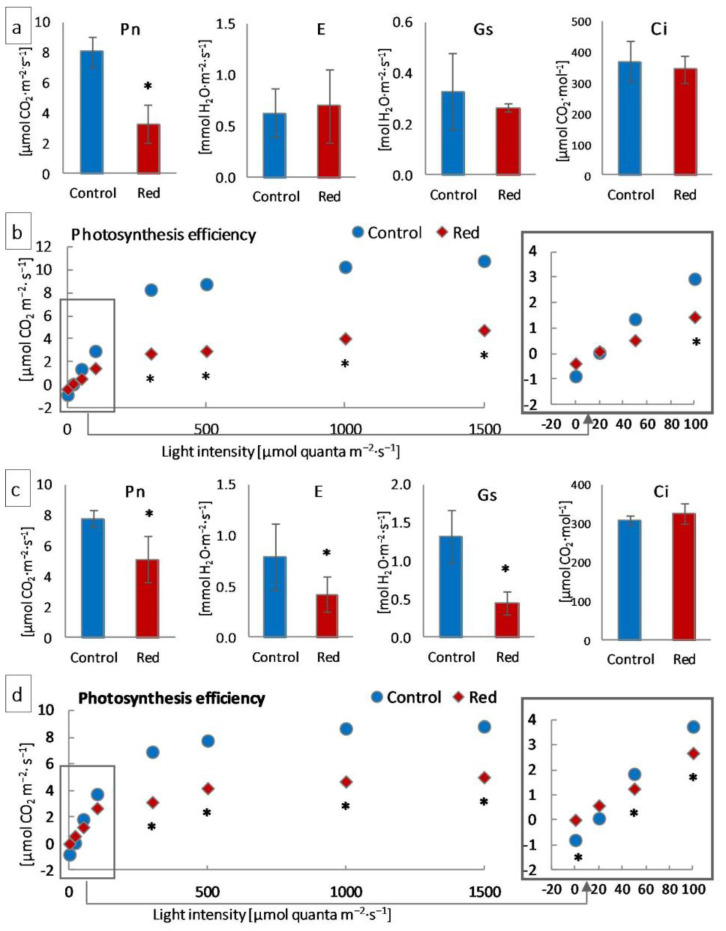
Photosynthesis efficiency of (**a**,**b**) butterhead and (**c**,**d**) iceberg lettuce types cultivated in transparent (control) and red glasshouses: (**a**,**c**) net photosynthesis (Pn), stomatal conductance (Gs), transpiration (E), and intercellular CO_2_ concentration (Ci) at 300 μmol quanta·m^−2^·s^−1^; (**b**,**d**) leaf and stem photosynthesis efficiency; * statistically significant differences within each parameter at *p* ≤ 0.05; (*n* = 3).

**Figure 4 cells-12-01552-f004:**
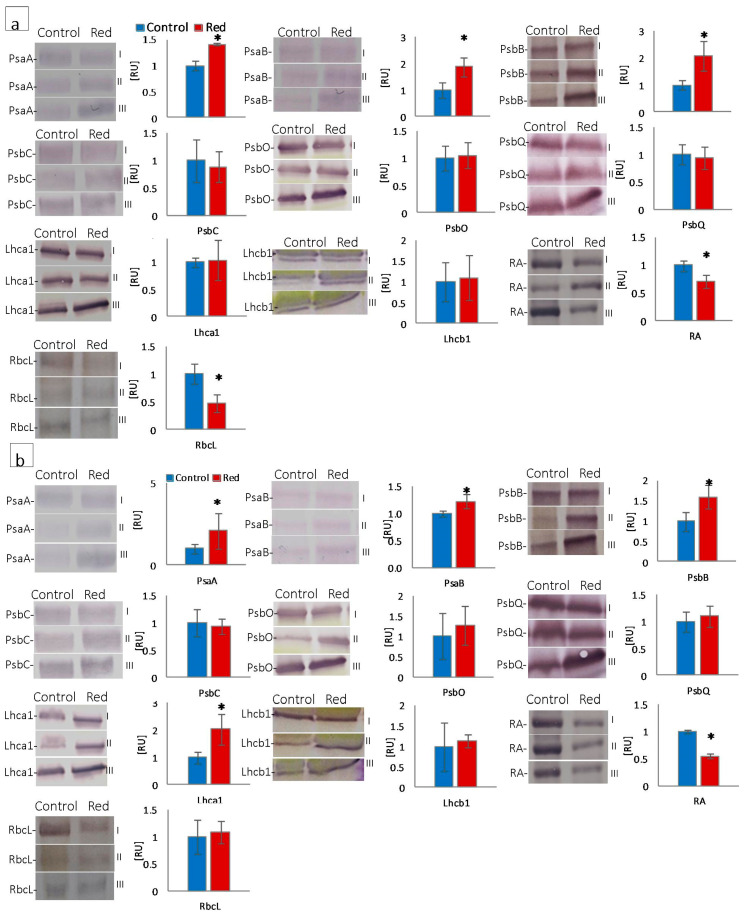
Structural and functional photosynthetic proteins content of (**a**) butterhead and (**b**) iceberg lettuce types cultivated in transparent (control) and red glasshouses; amount of protein loaded per lane—10 μg for Lhcb1, RbcL; 5 μg for PsaA, PsaB, PsbB, PsbC, PsbQ, Lhca1, RA; 3 μg for PsbO; content of proteins expressed as relative units (RU) in reference to the mean area value for the control on each gel expressed as 1; for the protein names, see Material and Methods (2.3.4); I, II, III—replications; * statistically significant differences within each parameter at *p* ≤ 0.05; (*n* = 3).

**Figure 5 cells-12-01552-f005:**
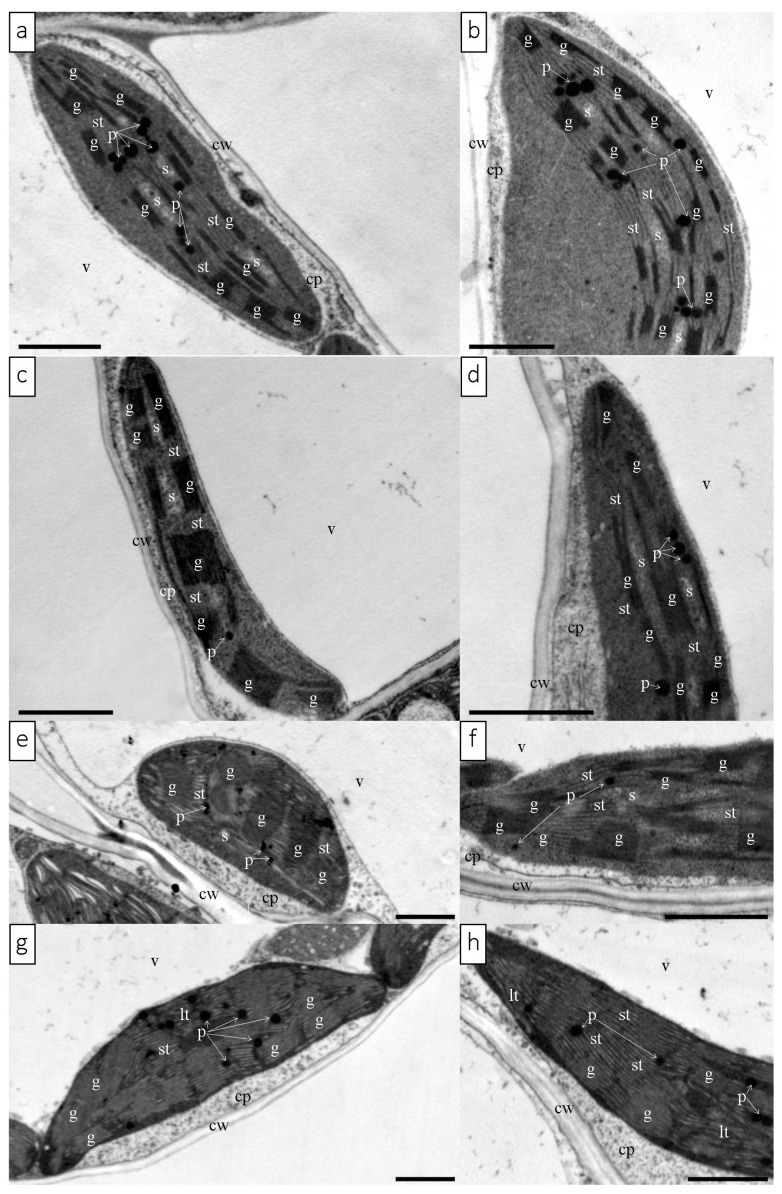
Chloroplast ultrastructure of butterhead (**a**,**d**) and iceberg (**e**,**h**) lettuce types cultivated in transparent (control) (**a**,**b**,**e**,**f**) and red (**c**,**d**,**g**,**h**) glasshouses. Abbreviations:, cp—cytoplasm; cw—cell wall, g—grana, lt—loose thylakoid; p—plastoglobuli, s—starch grain, st—stromal thylakoid, v—vacuole. Scale bars: 1 μm.

**Figure 6 cells-12-01552-f006:**
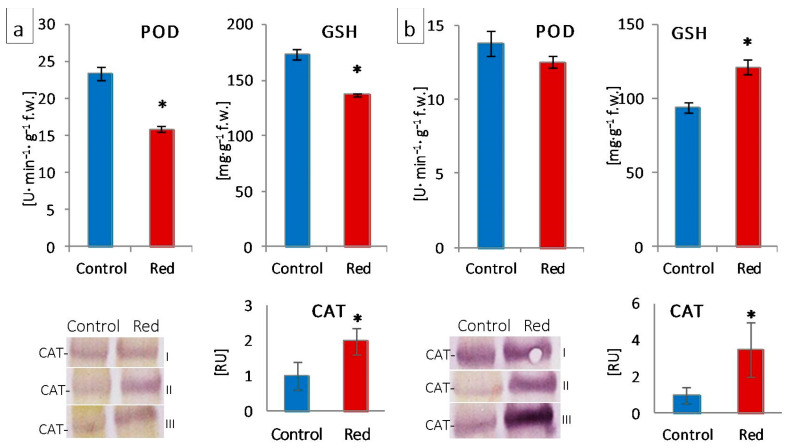
Activity of guaiacol peroxidase (POD) and content of glutathione (GSH) and catalase (CAT) of (**a**) butterhead and (**b**) iceberg lettuce types cultivated in transparent (control) and glasshouses; amount of protein loaded per lane—10 μg; content of proteins expressed as relative units (RU) in reference to the mean area value for the control on each gel expressed as 1; I, II, III—replications; * statistically significant differences at *p* ≤ 0.05; (*n* = 3).

**Table 1 cells-12-01552-t001:** Photosynthetic pigments’ concentration and ratio in leaves of butterhead and iceberg lettuce cultivated in transparent (control) and red glasshouses.

Lettuce Type	Conditions	Pigment Concentration ^a^ (mg·g^−1^ FW)				Pigment Ratio (RU)
		Chl *a*	Chl *b*	Chl *a + b*	Car	Chl *a*/*b*
Butterhead	Control	0.20 ± 0.03	0.06 ± 0.01	0.27 ± 0.04	0.07 ± 0.01	3.3 ± 0.1
	Red	0.28 * ± 0.03	0.08 * ± 0.01	0.35 * ± 0.03	0.07 ± 0.00	3.6 * ± 0.1
Iceberg	Control	0.26 ± 0.16	0.10 ± 0.04	0.36 ± 0.2	0.11 ± 0.05	2.4 ± 0.4
	Red	0.29 ± 0.07	0.11 ± 0.02	0.40 ± 0.09	0.09 ± 0.02	2.6 ± 0.2

^a^ Chl *a*—chlorophyll *a*; Chl *b*—chlorophyll *b*; Chl *a* + *b*—total chlorophylls; Car—carotenoids, RU—relative units. * Statistically significant difference within each parameter and lettuce type at *p* ≤ 0.05; (*n* = 5).

## Data Availability

The data are contained within this article.

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
