# Peer review of "Performance of the Photosynthetic Apparatus under Glass with a Luminophore Modifying Red-To-Far-Red-Light Ratio—A Case Study"

_cells, 2023, doi:10.3390/cells12111552_

Round 1
Reviewer 1 Report (New Reviewer)
The manuscript is recommended to be accepted.
Author Response
Thank you for comment
Reviewer 2 Report (Previous Reviewer 2)
Formal matters were improved, but two logical points were not revised.
(1) The second paragraph of Discussion
The correlation between the accumulation of proteins composing photosystems and chlorophyll a/b ratio should be discussed referring corresponding figures.
(2) The last paragraph of Discussion
A logical connection between citations concerning linear electron transport and results of this manuscript is unclear. Do you want to say ‘the red:far-red ratio modulates linear electron flow’ or not?
Author Response
Response to Reviewer 2 Comments
Formal matters were improved, but two logical points were not revised.
Point 1: (1) The second paragraph of Discussion.
The correlation between the accumulation of proteins composing photosystems and chlorophyll a/b ratio should be discussed referring corresponding figures.
Response 1: We complemented Discussion (see Lines: 429-431).
Point 2: (2) The last paragraph of Discussion
A logical connection between citations concerning linear electron transport and results of this manuscript is unclear. Do you want to say ‘the red:far-red ratio modulates linear electron flow’ or not?
Response 2: We complemented Discussion (see Lines: 470-471).
This manuscript is a resubmission of an earlier submission. The following is a list of the peer review reports and author responses from that submission.
Round 1
Reviewer 1 Report
The reviewed manuscript by Tokarz et al. titled “Ratio of blue to red or red to far-red light - what is more crucial for the efficient performance of the photosynthetic apparatus - a case study” concerns mainly of the influence of a red luminophore in glass of a greenhouse on the photosynthetic parameters in two lettuce varieties, as well as protein compositions and chloroplast structures. In my opinion, the title of the work does not connect directly with the main writing line of the study, in spite of the fact that the authors show at the beginning the differences in obtained spectra as well as indicate ratios between blue/red and red/far-red. The entire text is dedicated to the comparison of plants grown under usual glass and under glass containing red luminophore only.
I doubt the correct results of Western blot, because the authors made serious mistakes in temperature conditions. In addition, the authors often don’t take their attention in the connections between results in different experiments, which should support each other, but they don’t. Some interpretations are very brave. There are a lot of inaccuracies and descriptions.
Based on this I can not recommend the manuscript for publication in the current view.
I list my main comments below.
The title of the manuscript in the present view should be like “Influence of the red luminophore on….”.
L55. Chlorophyll and Carotenoid can be abbreviated here. In addition to PC, cytochrome c6 also acts there.
L99-105 After ‘furthermore’ should be transferred to the discussion.
I don’t understand why the authors write that the pigment content analyses were conducted according to Lichtenchtaler (L151-152), but below that calculation was made by using Wellburn’s equations (L158-159). So, which approach exactly was used.
L159. Chl a+b, but L162 chlorophyll. Why?
The authors write that SDS-phoresis was made at 4 ºC (L195), however the SDS can crystallize easily under such conditions. I know that gels after storage in a fridge should be warmed up to room temperature. In contrast blotting was made at room temperature (L201), while all manuals indicate the temperature equal to 4 ºC. BioRad even completes their cells with cold storage batteries. I’d like to ask the authors made some blots with conditions described by me to indicate that their results are correct.
In addition, I don’t find SDS in the loading buffer, which is usually used, only PVPP, but what is it? (L191)
The authors should indicate the content of the TBST buffer (pH, mM, etc.) (L202)
Statistical bars are absent on Fig. 3b,d. It should be added.
In Fig 2 Vj and Vi are higher in red, this means that more reaction centers are ‘closed’. As a consequence, the electron transfer is reduced, which can induce the decrease of the Rubisco activity (probably it can also be not enough increase of pH of stroma for Rubisco activation) as seen from Fig 3 b,d. The authors should discuss this.
Why do two subunits of PSI (PsaA and PsaB) have different behavior under growing in a red glasshouse? (L329-331). Should they both correlate with PSI content by the same way?
In figures of Western blot analysis (Fig 4) the control columns are shown with bars? How can the authors calculate this if the results can be compared only from the same membrane? I tried to analyze the pictures of membranes, but the authors did not indicate the name of the tracks.
Which statistical approach the authors used to find statistically significant differences, for example in Fig 3b (PsaA)? The same in other graphs. I’m not sure that the statistical calculations here were made correctly.
The quality of electron pictures is not good enough, especially in the case of butterhead. I’m not sure that such pictures reflect the correct data. Just compare a and e. The black asterisk (not asterix) in h is not a starch, this is a part of a torn piece.
The authors indicate ‘and visible starch grains (Figure 5a,b,e,f)’, but there is no starch in a.
The authors indicate that ‘and practically no grana were observed (Figure 5g,h)’, but grana are more visible in these pictures. Nevertheless, if accept this as true, how can the authors explain the higher content of the PSII core protein (PsbB) in this case? As well as how they also explain the absence of any changes in content of the water oxidizing complex proteins (PsbO-Q)?
L390. I do not agree with the authors about indistinct arrangement of thylakoids. If you look closely, there is the usual arrangement.
L411-412. More correctly is ‘open reaction centers’, because they are all active actually. CR -?
Reviewer 2 Report
Overview:
This manuscript draws attention to the importance of the red:far-red ratio for photosynthetic apparatus. Several insufficient descriptions should be improved.
Major comments:
Quantitative explanations are insufficient.
· Fig. 4 and 6
Please indicate the amount of applied sample per each lane in figure legends.
Please explain [RU] normalized by corresponding control sample in figure legends.
· Fig. 5 and L350-351
Images of other chloroplasts corresponding to panel (a)-(h) in supplemental figures and quantification of granal thylakoid, stromal thylakoid, or starch grains are necessary to claim ‘significant differences’. L357 ‘practically no starch grains’, but a large starch grain is shown in (h).
L347 ‘each organ (leaf-uppercase, stem-lowercase)’ Did you analyze stem of lettuce?
Discussion
Please refer exact panel or figure to discuss your results. It is difficult to distinguish between results of this work and the known observations.
Please revise the second paragraph from two viewpoints, separately describe. (1) Whether changes in the amount of chlorophyll a and b are consistent with those of proteins composing photosystems? (2) Whether changes in the accumulation of proteins can explain alteration of ultrastructure of chloroplast? L400-401 is suitable for the next paragraph.
L431-437 Please clarify that these citations explain which results in this work.
L439-440 ‘efficient linear electron transport’ is different from results and claim of this manuscript. I cannot understand the logic of this part of discussion.
Minor comments:
L32 CO2 ‘2’ lowercase
L327 RuBiSco à RuBisCO
L327 ‘enzyme’ à rubisco activase
L397 ‘gran’ à grana
L403 LHC should be defined at the first appearance.
L410 OEC, L412 PSII, L414 PQ, these abbreviations are previously spelled out in this manuscript.